# Additivity Effect on Properties of Cemented Ultra-Fine Tailings Backfill Containing Sodium Silicate and Calcium Chloride

Bingwen Wang [1], Su Gan [1,*], Lei Yang [1,*], Zhongqi Zhao [2], Zhao Wei [1,*] and Jiachen Wang [1]

[1] School of Energy and Mining Engineering, China University of Mining and Technology (Beijing), Beijing 100083, China; wbw@cumtb.edu.cn (B.W.); 2110110223@student.cumtb.edu.cn (J.W.)

[2] Shandong Gold Design Consulting Co., Ltd., Yantai 264003, China; danlihai@sd-gold.com

* Correspondence: bqt2000103040@student.cumtb.edu.cn (S.G.); bqt2100202036@student.cumtb.edu.cn (L.Y.); bqt2200101015@student.cumtb.edu.cn (Z.W.); Tel.: +86-457-19985734957 (S.G.); +86-457-17865932106 (L.Y.); +86-457-18332767183 (Z.W.)

**Abstract:** Tailings from gold mines gradually approach ultra-fine, making mine backfill costs higher and strength lower, which poses a serious threat to the safety of underground personnel and equipment. It is well known that suitable chemical admixtures can enhance the working properties of mortar materials. Therefore, in order to achieve the purpose of reducing the cost of ultra-fine tailings backfill and improving the working performance of ultra-fine tailings filling slurry, this paper provides a study on the effect of sodium silicate and calcium chloride on the properties of ultra-fine tailings cemented backfill materials. The results of experimental studies through rheology, strength, and microstructural tests, etc., showed that the optimal proportioning parameters of cementitious materials are 76.92% blast furnace slag, 19.24% carbide slag, and admixtures of 2.88% sodium silicate and 0.96% calcium chloride. The 3, 7, and 28-day uniaxial compressive strength of the ultra-fine tailings cemented paste backfill with the newly formulated blast furnace slag-based cementitious material increased by 124%, 142%, and 14%, respectively, compared to that of the ultra-fine tailings cemented paste backfill with the P. O42.5 cement. The setting time for ultra-fine tailings cemented backfill slurry is shortened by the addition of admixtures, and the shear stress of the slurry is correlated with the amount of hydration product generation and its formation of flocculating structure. Moreover, the cost of the newly prepared cementitious material is much lower than that of traditional cement, which lays a good foundation for the cemented filling of ultra-fine tailings.

**Keywords:** ultra-fine tailings; admixture; rheological properties; mechanical properties; microstructure

## 1. Introduction

Tailings are the solid mineral waste created by the natural dewatering of tailings slurry from beneficiation plants, where high output and low comprehensive utilization efficiency are the major characteristics [1]. This leads to a significant loss of resources and pollution of the ecosystem. In order to reduce the environmental problems from the tailings in surface stockpiles, many academics have suggested tailings as the primary backfill material in underground mined-out areas [2–4]. The filling body in mined-out areas can support overburden pressures, ensure that vegetation, farmland, and buildings on the surface are not destroyed, and ensure the safety of underground operators and equipment. At the same time, workers can recycle underground safety pillars, reducing the waste of mineral resources [5]. Consequently, the use of tailings to backfill the mined-out areas is one of the directions for the development of the back-filling method [6].

Although the back-filling method could solve many of the issues brought on by underground mining, it is unable to completely dispose of the enormous volume of tailings in surface stockpiles. In addition, comprehensive utilization of tailings is also one of the most effective ways to dispose of tailings. However, less than 40% of the tailings are utilized, especially gold tailings [5]. The problem of surface stockpiles of significant amounts of

gold tailings can be resolved through the development of innovative techniques for the comprehensive utilization of gold tailings. Researchers have investigated the use of tailings as building materials. Chen et al. [7] investigated the effect of making bricks using clay and raw materials including iron ore tailings and fly ash. Shettima et al. [8] pointed out that when the fraction of river sand substituted by iron ore tailings grew from 25% to 50%, 75%, and 100%, the workability of concrete decreased, but all the strength and modulus of elasticity data were higher than in traditional concrete. By separating and recovering quartz, feldspar, and gold elements from gold tailings samples, Chen et al. [9] found that they could make high-quality glass from the quartz concentrate and ceramics from the feldspar. Wang et al. [10] found that the ideal blending ratio for using gold ore tailings as raw siliceous material for cement manufacture is 5%, and the ideal calcination temperature is 1450 °C. Deng et al. [11] investigated the effect of the red mud-unburned ceramsite employed as an adsorbent to remove phosphate from swine water. As a result, previous studies [7–11] indicate that gold mine tailings can be used as fine sand for construction and raw materials for ceramics, brickmaking, and cement. However, due to the large amount of tailings surface stockpiles, when the coarser particles and some particles with higher silica content in the tailings are utilized, there still exists a portion of fine and ultra-fine tailings that cannot be utilized, which will cause great damage to the surface ecology. Therefore, combining the comprehensive utilization of tailings with underground mined-out areas backfill will be a good way to solve the surface stockpiling of tailings. The process flow diagram is displayed in Figure 1.

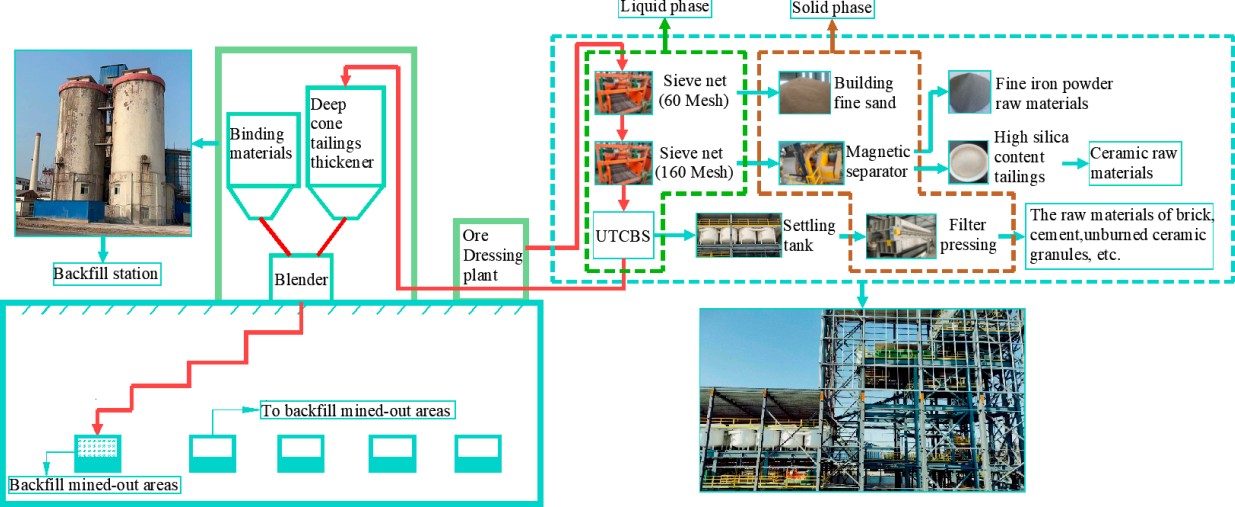

**Figure 1.** Process flow diagram of efficient comprehensive utilization and treatment of gold tailings.

From Figure 1, the tailings slurry from the beneficiation plant was first pumped to the tailings comprehensive utilization base, where it will be screened twice by meshes. The first level of screening uses a 60-mesh vibrating triple screen, and the products on the screen are sold as finished fine sand in concrete. The secondary screening uses a 160-mesh vibrating triple screen, and the products on the screen are transported to the magnetic separation apparatus through a belt. The magnetic material will be removed to improve the whiteness of the tailings. The tailings with high whiteness can be used as the primary raw material for ceramics and glass. Moreover, the removed iron-containing tailings can then be used as fine iron powder raw materials. After sedimentation and pressure filtration of the tailings slurry from secondary screening, the tailings can be used as a raw material for industrial unburned ceramic granules, bricks, cement, etc. The other portion of the tailings slurry can still be delivered to the mine backfill station, and the tailings deep-cone thickener can be used to raise the underflow concentration of the ultra-fine tailings (UT). Finally, the cementitious materials, water, and UT are mixed uniformly by the mixing system, and then

transported to mined-out areas by means of self-flow or pumping. So, all of the tailings in surface stockpiles can be effectively handled, and there is no need for a new tailings pond.

Tailings will become finer as beneficiation technology and comprehensive utilization of tailings technology advances [12–15]. Since ultra-fine tailings (UT) have the characteristics of slow particle settling, a unique chemical makeup, an uneven distribution of gradation, etc., they can be tough to treat [16]. Ke et al. [17,18] studied the effects of tailings fineness and gradation on the properties of the cemented paste backfill (CPB) and found that well-graded tailings may improve the CPB's capacity for consolidation. Portland cement is frequently used as the cementitious material to generate CPB. However, its production produces pollutants such as dust, $SO_2$, and $NO_2$, and the cost of buying cementitious material supplies accounts for roughly 70%–80% of the backfilling cost [19]. Therefore, establishing sustainable mine development requires a reduction in the cost of cementitious materials. After being physically or chemically activated, a few typical industrial wastes, such as blast furnace slag (BFS) (Appendix A), steel slag, carbide slag (CS), fly ash, etc., have potential activity and can be used as cementing materials [20]. Sun et al. [21] created a novel binder by using calcined quarry dust and NaOH at a mass ratio of 1:1 to activate slag, which significantly increased the uniaxial compressive strength (UCS) of the CPB. The rheological and mechanical properties of cemented backfill materials are influenced by factors such as the cement-tailings (C/T) ratio, mass concentration, curing temperature and time [22]. Underground mines will have different requirements for backfilling parameters in ultra-fine tailings cemented backfill slurry (UTCBS) because of specific surrounding rock characteristics, tailings gradation, and cost-effectiveness limits [23]. To meet particular requirements, some common admixtures (such as water reducers, pump agents, suspend agents, accelerators) are added to the backfilling materials [24,25]. Sodium silicate (SS) can be used as a gel filling in a variety of concrete applications because of its gelatinous characteristics. However, it is rarely used for backfilling underground mined-out areas [26]. The addition of SS will boost the early strength, speed up the hydration reaction, and lower the water-cement ratio of the cemented material to mine backfill [27]. However, when more SS is injected than a certain percentage, the strength of the cemented materials decreases [28]. There is disagreement about the mechanism by which SS affects the hydration of cementitious materials, even if some researchers concur that SS decreases the long-term mechanical strength of cemented materials [29]. Additionally, cemented backfill slurry (CBS) pipeline transportation is one of the vital components of underground mined-out areas backfilling. Guo et al. [30] examined the thixotropic properties of the UTCBS-added superplasticizer as well as the dynamic and static rheological properties. They also proposed a dynamic yield stress model that took into account the water content in flocs to enrich CBS rheological theory. Therefore, the use of the cementation filling method to treat UT will result in an increase in backfill cost, followed by the low strength of the filling body. It is necessary to carry out more research to enrich the theory of UT backfill and improve the application level of UT underground cemented backfill.

Due to the large surface stockpile of tailings and the high cost of backfilling ultra-fine tailings (UT), the technical difficulties are high. Most of the previous research focused on the unclassified tailings or classification tailings cementation backfill in the mined-out areas, with fewer studies on the ultra-fine tailings cementation backfill. Therefore, the main objectives of this study are to reduce the cost of backfilling UT and to improve the mechanical properties of the ultra-fine tailings cemented paste backfill (UTCPB). Firstly, suitable cementitious materials were prepared using blast furnace slag (BFS), carbide slag (CS), sodium silicate (SS), and calcium chloride (CC), to keep the cost of cementitious materials under control. Next, a slump test was used to ensure that the ultra-fine tailings cemented backfill slurry (UTCBS) met the pipeline delivery, and then the rheological test of the backfill slurry was carried out. Thirdly, on the basis of the newly prepared cementitious materials and the determined slurry proportioning parameters, mechanical experiments on the specimens of the UTCPB were carried out. Finally, the consolidation microscopic properties of the UTCPB were investigated by SEM-EDS, XRD, and mercury intrusion

porosimeter (MIP) tests. The research results of this thesis can promote the improvement of the theory and application level of UT cemented filling, and lay a good foundation for the realization of tailing-free mines.

## 2. Materials and Methods

### *2.1. Materials*

#### 2.1.1. UT and Cementitious Materials

Ultra-fine tailings (UT) were supplied by Shandong Gold Group Co. Ltd.'s Xincheng Gold Mine in China. A laser particle size analyzer was used to determine the tailings' grain size distribution, and the results are depicted in Figure 2. The grain size of the tailings spans 1.2 to 104.71 μm, with a coefficient of uniformity of 4.41 and a coefficient of curvature of 0.88, according to Figure 2 and Table 1. Wu et al. [31] defined the UT as an average particle size of less than 0.03 mm, a particle content of more than 50% below 0.019 mm, less than 10% above 0.074 mm, and less than 30% below 0.037 mm. Since the average particle size is 0.01 mm, the percentage of −0.019 mm particles is 72.4%, the percentage of +0.074 mm particles is 1.06%, and the percentage of +0.037 mm particles is 11.5%, and the tailings used in this experiment can be referred to as UT. Moreover, according to X-ray diffraction (XRD) analysis in Figure 3, the mineral compositions of UT are quartz, albite, and calcite. The main oxides in Table 2 are $SiO_2$, $Al_2O_3$, and CaO, which together account for more than 86 wt.% of the total.

The laser particle size analyzer and XRD were used to determine the mineral compositions and grain size compositions of the blast furnace slag (BFS) and carbide slag (CS) mixed cementitious materials, respectively, and the results of the tests and analyses are shown in Figures 2b and 3. The particle size range of mixed cementitious materials is 0.48–630.96 μm, with a median diameter of 11.48 μm and a specific surface area of 1.3 $m^2/g$, as can be seen in Figure 2b and Table 1. Moreover, from Figure 3a and Table 2, BFS is primarily composed of the minerals: quartz, calcite, melilite, and anorthite. It also contains significant amounts of CaO, $SiO_2$, and $Al_2O_3$, which together make up 83 wt.% of BFS. Calcite and portlandite make up the majority of CS, while CaO accounts for 87.45% of its total content.

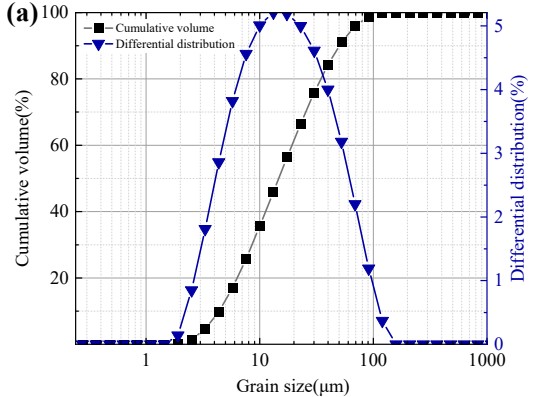
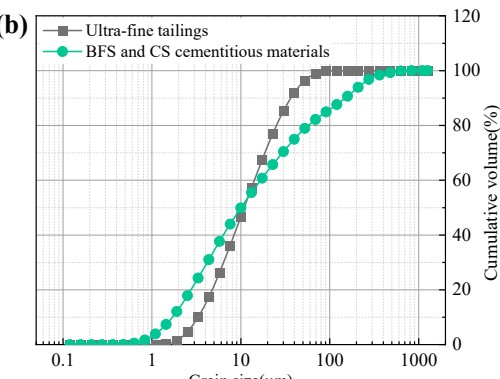

**Figure 2.** Grain size distribution of raw materials: (**a**) the cumulative and incremental curves of UT, and (**b**) the cumulative curves of UT and BFS and CS mixed cementitious materials.

**Table 1.** Basic physical characteristics of UT and mixed cementitious materials.

| Parameter | Cc | Cu | $D_{10}$ | $D_{30}$ | $D_{50}$ | $D_{60}$ | $D_{90}$ | Ss ($m^2/g$) |
|---|---|---|---|---|---|---|---|---|
| UT | 0.76 | 3.98 | 3.32 | 5.75 | 10.92 | 13.18 | 36.44 | 0.8 |
| BFS and CS | 0.58 | 9.12 | 1.91 | 4.34 | 11.48 | 17.38 | 158.49 | 1.28 |

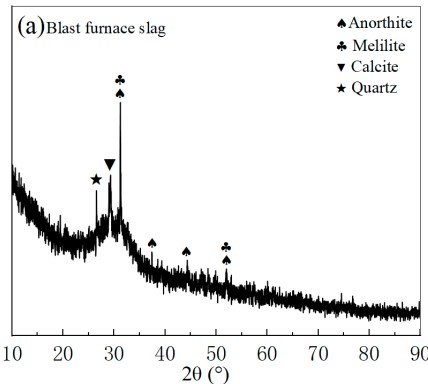
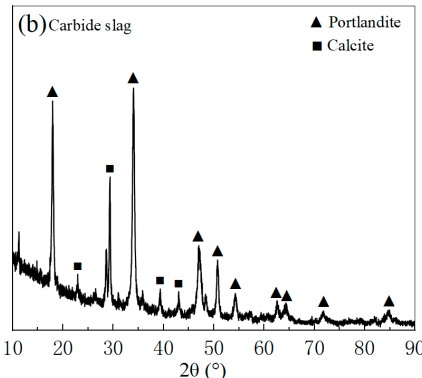

**Figure 3.** X-ray diffraction spectra of the raw materials: (**a**) BFS and (**b**) CS.

**Table 2.** Major chemical components of the UT and cementitious materials.

| Content (wt.%) | SiO₂ | Al₂O₃ | K₂O | CaO | Na₂O | Fe₂O₃ | MgO | SO₃ | TiO₂ | MnO | Other |
|---|---|---|---|---|---|---|---|---|---|---|---|
| UT | 64.97 | 18.29 | 5.64 | 3.17 | 2.84 | 2.47 | 0.93 | 0.44 | 0.33 | - | 0.92 |
| BFS | 28.51 | 16.63 | - | 37.85 | 1.4 | 1.52 | 8.9 | 2.31 | 0.78 | 0.57 | 1.53 |
| CS | 5.29 | 2.66 | - | 87.45 | 0.77 | 0.64 | 1.06 | 1.38 | 0.19 | 0.09 | 0.47 |

### 2.1.2. Admixtures SS and CC

The type of admixture and dose must be carefully chosen, because the choice of admixture has a significant impact on the hydration reaction process of the cementitious materials. To reduce transportation costs, Shandong Laizhou Fuli Sodium Silicate Co., Ltd., located near the mine, provided the sodium silicate (SS) used in this experiment. Table 3 displays the basic indicators of SS. Calcium chloride (CC) was obtained from Zhejiang Quzhou Jumbo Plastic Chemical Co., Ltd.'s industrial anhydrous CC. Among them, the content of CC is over 94%.

**Table 3.** Some basic indicators of SS.

| Appearance | SiO₂/% | Na₂O/% | Modulus | Density | Reference Standards |
|---|---|---|---|---|---|
| liquid | 28.5 | 9.07 | 3.24 | 1.391 | GB/T 4209-2022 |

### 2.1.3. Water

To obtain fresh mixtures, tap water was used as the mixing liquid. This tap water's main mineral constituents are Ca (43.70 ppm), Mg (2.35 ppm), and Na (3.10 ppm), and its PH level is 7.35.

### 2.1.4. Preparation of UTCBS and UTCPB

After being separately weighed by the experimental design, the following experimental materials were added to a mortar mixer: ultra-fine tailings (UT), blast furnace slag (BFS), carbide slag (CS), sodium silicate (SS), calcium chloride (CC), and water. The mixture was thoroughly stirred for 5 min while being carefully watched to ensure that there was no dry mixture present. It is necessary to pre-mix the dry mixture (UT, BFS, and CS) in the stirring pot using a scoop. Admixtures SS and CC were added to new tap water first, and then they were gradually added to the stirring pot. A single rotational speed (140 + 2 rpm) was used for the stirring procedure.

The prepared ultra-fine tailings cemented backfill slurry (UTCBS) was poured into two overlapped cubic (70.7 × 70.7 × 70.7 mm) molds, which were then held for two hours to allow tailings to settle. Before removing the top mold, excess backfill material was gently scraped away with a scraper. These ultra-fine tailings cemented paste backfill (UTCPB) specimens were stored at room temperature (20 °C ± 5) for 22 h and then placed in a

constant temperature and humidity curing box. The box was maintained at 20 °C ± 2 with a relative humidity of 90% to better simulate the field conditions in the underground mined-out areas. In order to test the effect of admixtures on the rheological and consolidation properties of UTCBS, experiments were designed with different cement-tailings (C/T) ratios, mass concentrations, and admixture dosages, respectively. The specific parameters for each component of the cementitious materials are shown in Table 4. It should be noted that the mass ratio of CS to BFS is 1/4, which remained constant, and the mass ratio of CC to the total mass ratio of BFS and CS is 1%, which remained constant. In CM4, CM5, and CM5, the mass ratio of CC to SS is 1, 1/2, and 1/3, respectively. The experimental design is shown in Table 5.

**Table 4.** Specific parameters for each component of the cementitious materials.

| No | BFS | CS | SS | CC | P. O32.5 Cement | P. O42.5 Cement |
|---|---|---|---|---|---|---|
| CM1 | - | - | - | - | 100% | - |
| CM2 | - | - | - | - | - | 100% |
| CM3 | 80% | 20% | - | - | - | - |
| CM4 | 78.44% | 19.6% | 0.98% | 0.98% | - | - |
| CM5 | 77.67% | 19.42% | 1.94% | 0.97% | - | - |
| CM6 | 76.92% | 19.24% | 2.88% | 0.96% | - | - |

**Table 5.** Design of laboratory tests on UT cemented backfill materials.

| Experimental Projects | Cementitious Materials | C/T Ratio [(A)] | Mass Concentration [(B)] | Curing Age (Days) |
|---|---|---|---|---|
| Setting time | CM3, CM4, CM5, CM6 | 1/4, 1/6, 1/8, 1/10 | 63%, 61%, 59%, 57% | - |
| rheology | CM3, CM4, CM5, CM6 | 1/4 | 63%, 61%, 59%, 57% | - |
| UCS | CM1, CM2, CM3, CM4, CM5, CM6 | 1/4, 1/6, 1/8 | 63%, 61%, 59%, 57% | 3, 7, 28 |
| SEM | CM3, CM6 | 1/4 | 63% | 7 |
| SEM (net paste) | CM6 | - | 63% | 7 |
| XRD (net paste) | CM3, CM6 | - | 63% | 3, 7 |
| MIP | CM3, CM6 | 1/4 | 63% | 7 |

$$(A) = \frac{M_{cementious\ materials}}{M_{tailings}}, (B) = \frac{M_{tailings} + M_{cementious\ materials}}{M_{tailings} + M_{cementious\ materials} + M_{water}}.$$

### 2.2. Methods

### 2.2.1. Fluidity and Setting Time Tests

Slump is a workability indicator for conventional concrete, and this study also uses it to assess ultra-fine tailings cemented backfill slurry (UTCBS)'s workability. The curtailed cone has a diameter of 200 mm at the base, 100 mm at the top, and a height of 300 mm. Typically, UTCBS was filled three times to the curtailed cone, with each application having a 100 mm fill height and being tamped 25 times with a tamping stick. Then, the UTCBS was filled to a level above the truncated cone's rim, and any excess UTCBS was scraped off with a scraper. At last, the constrained cone is quickly raised, leaving UTCBS on a rigid base plate that is horizontal and falls due to gravity. The slump test is conducted by Chinese Standard GB/T 50080-2002. A slump value of 230 mm or more was considered to meet transport requirements in the studied gold mine. UTCBS was created using A6 cementitious material (as shown in Table 4), ultra-fine tailings (UT), and water, with a mass concentration of 63% and a cement-tailings (C/T) ratio of 1:4. The maximum mass concentration in the present research was set at 63% since its slump value was experimentally determined to be 233 mm.

While a longer setting time would delay the mining and backfill cycles, a shorter setting time would cause the UTCBS to obstruct the pipeline during the transport operation. As a result, one of the crucial things to consider when assessing the quality of mined-out areas backfill is the UTCBS setting time. The Vicat apparatus is utilized in this investigation to determine the UTCBS setting time. Every 5 and 15 min, as the UTCBS approaches the setting state, an experiment is run to precisely determine the setting time of the UTCBS.

2.2.2. Rheological and Mechanical Properties Tests

Ultra-fine tailings cemented backfill slurry (UTCBS)'s rheological behavior was investigated using a computer-controlled rotational rheometer, Rheolab QC (Anton Paar, Graz, Austria), equipped with a blade rotor's type, ST34-2D/2V/2V-30/156. The ASTM C1749-2012 standard approach was used to assess rheological characteristics. In this experiment, two types of measurements were set up. The first is to keep the rotor speed constant and study the rheological properties of the ultra-fine tailings cemented paste backfill (UTCPB) at five rotational speeds: $20\,\mathrm{s}^{-1}$, $40\,\mathrm{s}^{-1}$, $60\,\mathrm{s}^{-1}$, $80\,\mathrm{s}^{-1}$, and $100\,\mathrm{s}^{-1}$, respectively. The second one is to set the rotating speed of the rotor to increase gradually from $0\,\mathrm{s}^{-1}$ to $120\,\mathrm{s}^{-1}$ and to study the rheological properties of the slurry at different mass concentrations and holding times.

Uniaxial compressive strength (UCS) is one of the macroscopic features of ultra-fine tailings cemented paste backfill (UTCPB) consolidation and has a significant impact on the stability of the surrounding rock. When the samples reach the curing time (3, 7, and 28 days), the HDS-50 UCS testing machine were used to evaluate the UCS of three UTCPB samples. For each proportioning parameter, three specimens of the filling body were tested, and the average value was selected as the UCS value of the filling body for the proportioning parameter. All of these experimental procedures were carried out according to the JGJ/T70-2009 standard. The HDS-50 UCS testing machine has a 50 kN loading capacity, and each sample was loaded at a constant rate of 0.5 mm/min.

2.2.3. Microstructural Characterization

Scanning electron microscopy (SEM), X-ray energy dispersive spectrometry (EDS), mercury intrusion porosimeter (MIP), diffraction (XRD), and X-ray were used to observe the specimens' microstructure. The internal structure of the ultra-fine tailings cemented paste backfill (UTCPB)'s pores, the hydration products' morphology, and the connections between the material's particles are all visible via SEM. In order to understand the elemental and mineralogical composition of each morphological structure, EDS was utilized to measure the elements present in a small area of the surface of a microstructure in a UTCPB specimen. The pore structure of UTCPB specimens was examined using MIP, including pore size, distribution, and volume. The relevant criteria for specimen preparation and operating procedures were strictly followed during the execution of SEM, EDS, XRD, and MIP experiments. The ZEISS GeminiSEM 300, X'Pert PRO MPD, and AutoPore V 9620 were the test apparatus models for SEM-EDS, XRD, and MIP, respectively.

## 3. Results and Discussion

### 3.1. Effect of Admixture Dosages, Mass Concentrations, and C/T Ratios on UTCBS's Setting Time Tests

The initial and final setting times of ultra-fine tailings cemented backfill slurry (UTCBS) are depicted in Figure 4 as a function of the admixture dosages, mass concentration, and cement-tailings (C/T) ratio. From Figure 4a, while maintaining the quantity of calcium chloride (CC) added, the initial and final setting times of UTCBS were shortened with the addition of more sodium silicate (SS). As described in Section 3.4.1 by microscopic characteristics, the addition of SS produces more calcium silicate, which fills and shortens the spaces between the ultra-fine tailings (UT) particles. It is obvious from all of the samples that the UTCBS samples' setting times lengthen as the mass concentration falls. This is primarily because adding more water will cause the distance between slurry particles to expand, making it difficult for the internal mesh structure of the ultra-fine tailings cemented paste backfill (UTCPB) to form. Similar to how UTCBS's initial and final setting times rise as the C/T ratio drops. This phenomenon can be explained by the fact that less cementitious material means fewer hydration products, which is bad for the coagulation and solidification of the interior structure. Visually, we can see from Figure 4 that UTCBS's C/T ratios have a greater impact on the initial and final setting times of the slurry than UTCBS's mass concentration. The initial and final setting times apart increased by 456 min

and 600 min as the slurry mass concentration decreased from 63% to 57%, but apart increased by 1128 min and 1670 min as the slurry C/T ratio decreased from 1/4 to 1/10.

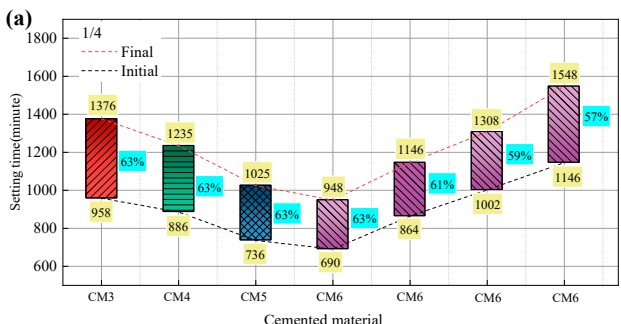
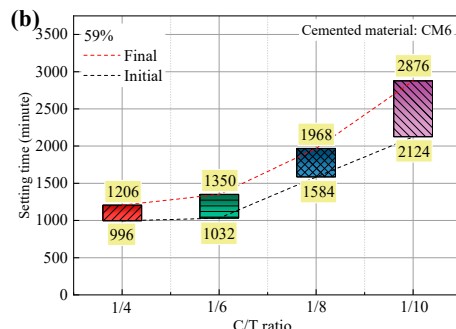

**Figure 4.** Effect of different influencing factors on the setting time of UTCBS: (**a**) admixture dosage and mass concentration; and (**b**) C/T ratios.

### 3.2. Effect of Admixture Dosages, Steady State Shear Rates, Mass Concentrations, and Holding Times on UTCBS's Rheological Behavior

3.2.1. Mechanisms of Flocculation Structure Formation and Development in UTCBS

The ultra-fine tailings (UT) particles produce a double electric layer structure as a result of the chemicals' flotation action in the beneficiation plant, which will absorb the cations in the ultra-fine tailings cemented backfill slurry (UTCBS). The cations closer to the tailing particles are attracted more, and the cations further from the tailing sand particles are attracted less under the influence of the static stress field. The water in this electrostatic structure is known as bound water or adsorbed water film. At the same time, polar water molecules are also trapped in the electrostatic structure because of the gravitational field. The continuous stirring of the mixture during the making of the UTCBS causes the adsorbed water film on the surface of each tailings particle to transform into a common water film. Which in turn can connect numerous tailings particles together and lead to the formation of a flocculating structure of the UTCBS [32]. When UTCBS started to mix, the cementitious elements inside it started to undergo a hydration reaction. Gradually, a small amount of hydration products like amorphous gel, calcium hydroxide, ettringite, and other hydration products have been produced. The development process of the flocculation structure of UTCBS is shown in Figure 5.

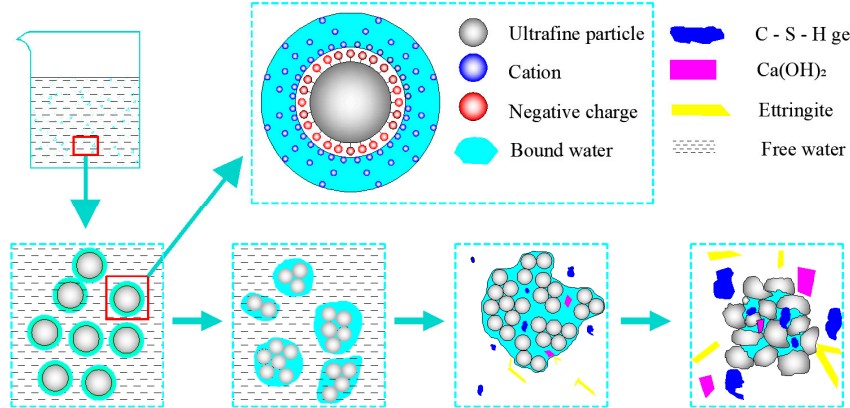

**Figure 5.** Schematic diagram of flocculation structure formation and development of UTCBS.

3.2.2. Effect of Admixture Dosages and Steady State Shear Rates on the Rheological Properties of UTCBS

Figure 6 displays the variation curves of ultra-fine tailings cemented backfill slurry (UTCBS) shear stress with time at various steady-state shear rates. Figure 6a shows that the initial shear stress without admixture is 757 Pa at a steady state shear time of 800 s and

998 Pa at mass concentrations of 63% UTCBS and 1/4 cement and tailings. When sodium silicate (SS) and calcium chloride (CC) were introduced as 1% and 1%, 2% and 1%, and 3% and 1%, the initial shear stresses were 1140 Pa, 1270 Pa, and 1580 Pa, respectively; and when the steady state shear time was 800 s, the shear stresses became 814 Pa, 926 Pa, and 1074 Pa, respectively. This indicates that the addition of SS and CC increased the shear stress of UTCBS, which was subsequently followed by an initial quick decline in the shear stress-time curve. Then, as the shear time increases, the rate of decrease in shear stress begins to slow down. It is a typical shear thinning or pseudoplastic behavior [33]. Second, the presence of CC and SS caused a significant amount of $Cl^-$, $Ca^{2+}$, and the initial slurry-charged tailings particles to adsorb together. At this point, CC and SS undergo chemical reactions, including ionic reactions, around the tailings particles to produce calcium hydroxide and calcium silicate precipitates. This is a cementation phenomenon rather than solidification [34], so it increases the cohesiveness of the UTCBS and the envelopment of the tailings particles.

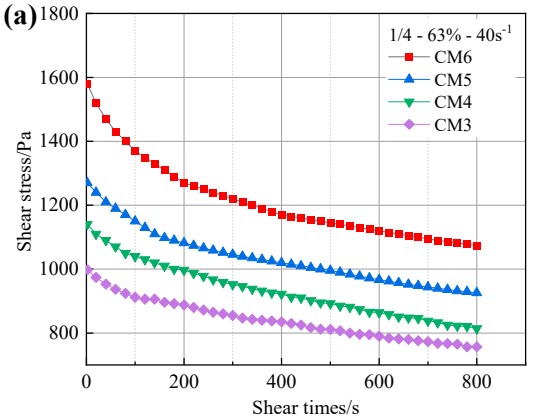
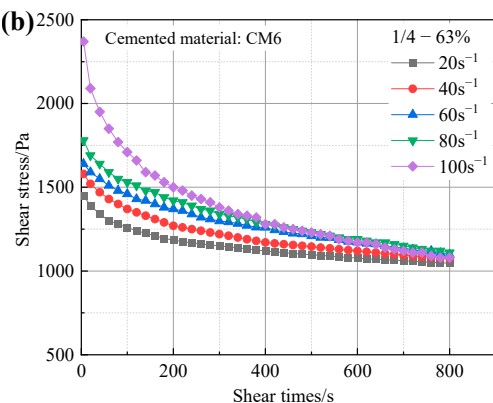

**Figure 6.** Change in shear stress of UTCBS as a function of shear time under different influencing factors: (**a**) admixture dosages; and (**b**) steady state shear rates.

Figure 6b shows that the initial shear stresses increase as the steady state shear rate increases. This is because the momentum law f = mv, which states that the shear stress is equal to the product of the mass of the slurry particles around the rotating rheometer rotor and the tangential velocity. So, the initial shear stress is greater as the state shear rate increases. Due to the ultra-fine tailings cemented backfill slurry (UTCBS)'s inherent "shear thinning" quality, the slurry shear stress will continue to decrease as the steady state shear action is there. In addition, the UTCBS system can be significantly impacted by a variety of microstructures, hydrodynamic forces, Brownian motions, and colloidal forces of the particles inside these microstructures [35]. The initial shear stress of the slurry corresponds to the start-up pressure of the slurry pipeline, which is essential for the selection of pumping equipment.

### 3.2.3. Effect of Mass Concentrations and Holding Times on the Rheological Properties of UTCBS at Continuously Varying Shear Rates

The change in shear stress of ultra-fine tailings cemented backfill slurry (UTCBS) as a function of shear rate for various mass concentrations and holding times were shown in Figure 7. According to Figure 7a, the rheological curves of various concentrations of UTCBS exhibit a three-stage growth trend. The three stages are rapid growth, slow decrease, and slow growth. For instance, the middle portion of the rheological curves for mass concentrations of 61% and 63% had obvious negative slopes when the shear rate was in the phases of 10–20 $s^{-1}$ and 20–60 $s^{-1}$, respectively. However, the slope thereafter changes to positive, despite its tiny value. This phenomenon arises from the fact that slurries with higher mass concentrations have more hydration products and a higher development of flocculating structures. As the rotor shear rate gradually increases, the first stage of shear stress increases rapidly. The middle portion of the curve had a negative slope due to the

floc structure produced in the stage of slurry preparation being damaged by rotor shear action. The destroyed floc structure reduced the shear stress of the slurry. At the third stage, the destruction and rebuilding of the slurry structure had essentially reached equilibrium, and the shear stress of the slurry depended mostly on the rise in shear rate.

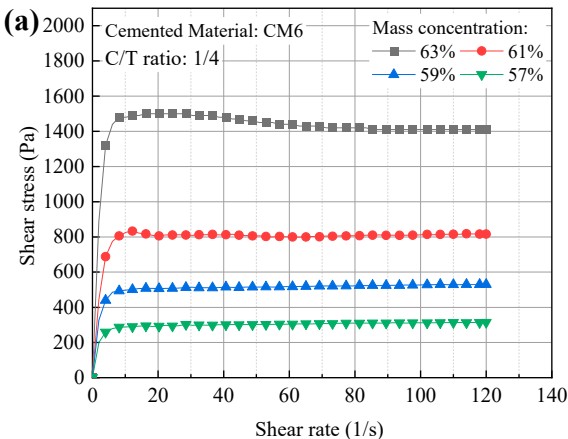
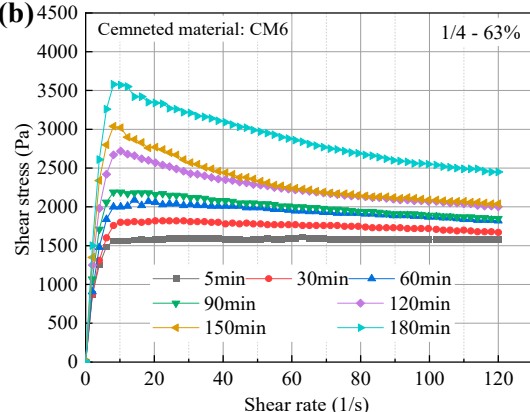

**Figure 7.** Change in shear stress of UTCBS as a function of shear rate under different influencing factors: (**a**) mass concentrations; and (**b**) holding times.

According to Figure 7b, freshly prepared and hosted ultra-fine tailings cemented backfill slurry (UTCBS) have some similar rheological properties. The maximum shear stresses were 1610, 1820, 2090, 2190, 2720, 3040, and 3580 Pa when the holding time of the filled slurry was 5, 30, 60, 90, 120, 150, and 180 min, respectively; and the corresponding shear rates were 63.1 $s^{-1}$, 28.5 $s^{-1}$, 14.2 $s^{-1}$, 10.2 $s^{-1}$, 10.2 $s^{-1}$, 8.13 $s^{-1}$. The shear stresses were 1580, 1670, 1820, 1850, 2000, 2040, and 2450 Pa, respectively, when the shear rate was 120 $s^{-1}$. It is shown that when the slurry's holding duration grows, the maximum shear stress of the UTCBS increases. It is also possible to find that the first stage of the curve shortens, the second stage lengthens, and the third stage shortens. This is due to the cementitious material's hydration reaction occurring during the holding period in CUTBS, which led to the construction of a network structure between the tailing particles. This structure is different from the flocculation structure mentioned in Section 3.4.1.

### 3.3. Effect of Types of Cementitious Material, Admixture Dosages, Mass Concentrations, and C/T Ratios on UTCBS's Rheological Behavior

Ultra-fine tailings cemented paste backfill (UTCPB) specimens of cement and four various ratios of blast furnace slag (BFS)-based cementitious materials were made in a lab and examined for their uniaxial compressive strength (UCS). This provided a comparison of the consolidation of ultra-fine tailings (UT) by cement and newly prepared cementitious materials, and ascertained the impact of sodium silicate (SS) and calcium chloride (CC) on the UTCPB. The effects of admixture dosages, mass concentration, and cement-tailings (C/T) ratios on the UCS of UTCPB specimens are shown in Figure 8. Compared to the ultra-fine tailings cemented paste backfill with the P. O42.5 cement, the 3, 7, and 28-day uniaxial compressive strengths of the ultra-fine tailings cemented paste backfill with the newly formulated blast furnace slag-based cementitious material increased by 124%, 142%, and 14%, respectively. It is clear that the UCS of UTCPB specimens prepared from BFS-based cementitious materials with added admixtures is much higher than that of P. O32.5 and P. O42.5 cement. UTCPB specimens prepared from BFS-based cementitious materials without additives had UCS below P. O42.5 cement for only 3 days of curing age. This demonstrates that BFS has a good consolidation effect on UT. The UCS of the UTCPB of BFS-based cementitious materials with SS and CC is higher than that without admixture, especially the short-term UCS, as can be seen in Figure 8a. When the curing period is 3 days, the admixture amount of SS is 3%, 2%, and 1%, and the UCS is increased by 159%,

138%, and 103%, respectively, but the UCS is only increased by 9.4%, 6%, and 1.7% when the curing time is 28 days. Therefore, the optimum additive ratio is 3% sodium silicate and 1% calcium chloride. The optimum ratio of the newly prepared cementitious material was calculated as 76.92% blast furnace slag, 19.24% carbide slag, and admixtures of 2.88% sodium silicate and 0.96% calcium chloride. This shows agreement with the findings of Guo et al. [36] about UCS growth trends. However, the latter group concluded that adding CC causes the short-term strength growth of lead-zinc smelting slag-based cemented tailings backfill to be low and the long-term strength growth to be high. The findings of this study show that adding SS and CC causes the UCS of BFS-based UTCPB to grow strongly over the short term but slowly over the long term. Specific analyses of reasons for this were depicted in Section 3.4.

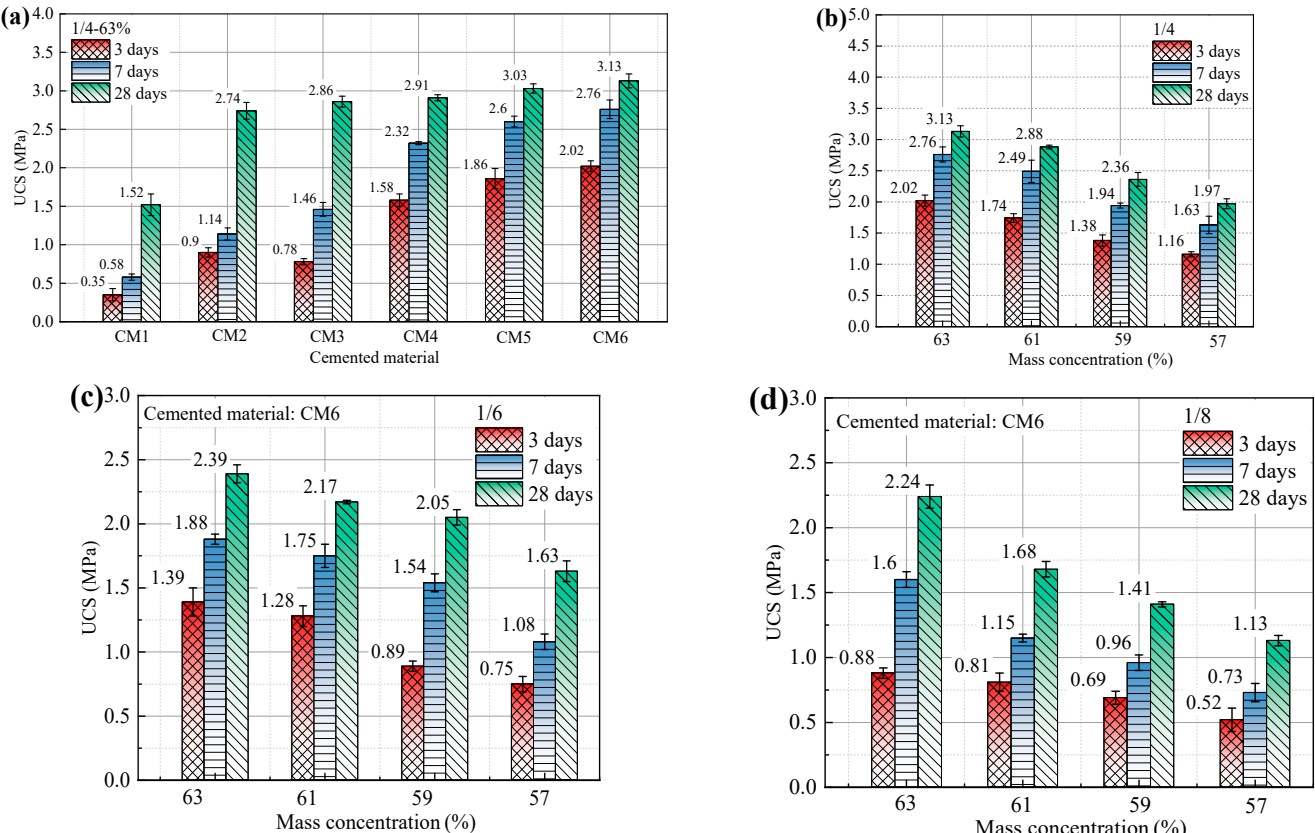

**Figure 8.** Effect of different influencing factors on the UCS of UTCPB specimens: (**a**) types of cementitious material and admixture dosages; (**b**) 1/4 ratio and different mass concentration; (**c**) 1/6 ratio and different mass concentration; (**d**) 1/8 ratio and different mass concentration.

As shown in Figure 8b–d, the uniaxial compressive strength (UCS) of the ultra-fine tailings cemented paste backfill (UTCPB) falls when the cement-tailings (C/T) ratio of the ultra-fine tailings cemented backfill slurry (UTCBS) is reduced. The increment between short-term and long-term strength increases with a decreasing C/T ratio. For instance, when the C/T ratio is 1/4, 1/6, or 1/8 and the mass concentration of UTCBS is 63%, the UCS of the UTCPB rises by 0.37, 0.51, and 0.64 MPa in 7–28 days, respectively. Additionally, for the same C/T ratio, the UCS of the UTCPB declined almost linearly as the mass concentration of the UTCBS decreased. This demonstrates that water plays a significant role in the UTCBS's consolidation properties. As a result of the ultra-fine tailing (UT) particles' fineness, high specific surface area, high water retention, and need for more cementitious materials, mass concentration has a stronger impact on the UTCPB.

### 3.4. Hydration Mechanism of Cementitious Materials and Microstructural Analyses of UTCPB Specimens

### 3.4.1. XRD and Hydration Mechanism

Figure 9 displays the scanning electron microscopy (SEM) image and XRD spectrum of ultra-fine tailings cemented paste backfill (UTCPB) net paste samples with and without admixtures that were cured in a constant temperature and humidity curing box for 3 and 7 days, respectively. According to Equation (1) [37] and Table 2, the content of CaO in carbide slag (CS) is up to 87.45%, which offers a sufficiently alkaline environment for the ultra-fine tailings cemented backfill slurry (UTCBS). When blast furnace slag (BFS) and CS are dissolved in water, the strong alkaline environment stimulates the activity of the BFS, prompting the activation and decomposition of the BFS vitreous particles and the depolymerization reaction. At this point, the slurry system will produce a significant amount of free $Ca^{2+}$, $SiO_4^{4-}$, and $AlO_4^{5-}$. According to Equations (2)–(4), $Ca^{2+}$ adsorbed with $SiO_4^{4-}$ to produce C-S-H gels and $Ca^{2+}$ adsorbed with $AlO_4^{5-}$ to produce C-A-H gels, which led to C-A-S-H gels as Ca/Si increased [38,39]. C-S-H denotes hydrated calcium silicate; C-A-H denotes hydrated calcium aluminates; and C-A-S-H denotes hydrated calcium aluminosilicate. In general, C, A, S, and H can be used to denote CaO, $Al_2O_3$, $SiO_2$, and $H_2O$ in the chemical formula, respectively [39].

$$CaO + H_2O \rightarrow Ca\,(OH)_2 \tag{1}$$

$$SiO_2 + Ca(OH)_2 + H_2O \rightarrow \text{C-S-H} \tag{2}$$

$$Al_2O_3 + Ca(OH)_2 + H_2O \rightarrow \text{C-A-H} \tag{3}$$

$$Na_2O \cdot Al_2O_3 \cdot 6SiO_2 + Ca\,(OH)_2 + H_2O \rightarrow \text{C-(A)-S-H} \tag{4}$$

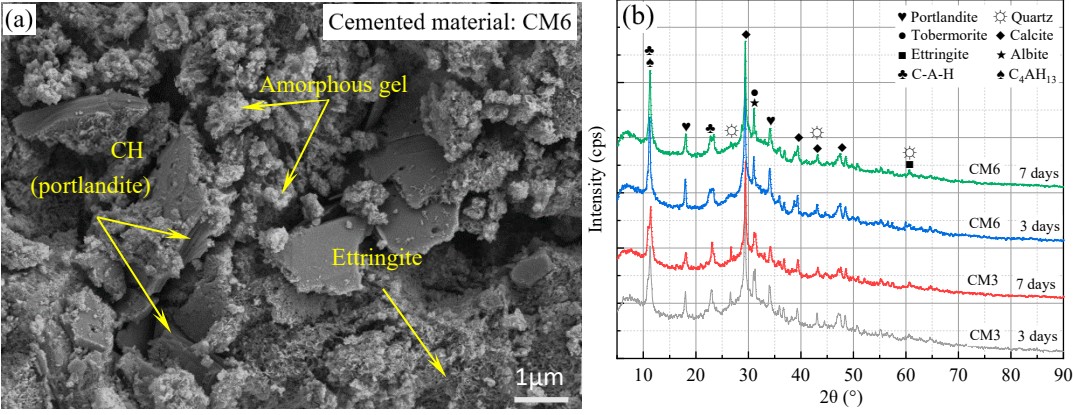

**Figure 9.** Micro testing of net paste samples of BFS-based cementitious materials: (**a**) SEM tests, and (**b**) XRD tests.

Stojanovia and Vajgand [40] used continuous titration calcium chloride (CC) solutions and a standard solution of sodium silicate (SS) to investigate the formation mechanism of calcium silicate compounds. It can be represented by the general formula $xCaO\text{-}SiO_2$. Equation (5) is the fundamental chemical equation for the reaction between SS and CC.

$$CaCl_2 + Na_2SiO_3 = CaO \cdot SiO_2 + 2NaCl \tag{5}$$

The diffraction apex of the aluminosilicate mineral anorthite was observed in Figure 9b, and one of its minerals, C3A, underwent a hydration reaction to produce $C_4AH_{19}$ and $C_2AH_8$ in an alkaline solution according to Equation (6) [41]. At room temperature, $C_4AH_{19}$

and $C_2AH_8$ are hexagonal plate crystals in a metastable state that are easy to change into other crystals like $C_3AH_6$, as shown in Equation (7) [41].

$$2\,C_3A + 27\,H \rightarrow C_4AH_{19} + C_2AH_8 \tag{6}$$

$$C_4AH_{19} + C_2AH_8 \rightarrow 2\,C_3AH_6 + 9\,H \tag{7}$$

The blast furnace slag (BFS) vitreous is further dissociated in a strongly alkaline environment, where a small quantity of $SO_3$ readily reacts with water to create $SiO_4^{2-}$. The solution's $Ca^+$ and $SiO_4^{2-}$ will also combine with $C_3AH_6$ to create ettringite. As a result, minor amounts of ettringite can be seen in the SEM picture (Figure 9a) and XRD spectrum (Figure 9b). Equations (8)–(10) demonstrate the three reaction equations.

$$C_3A + CH + 12\,H \rightarrow C_4AH_{13} \tag{8}$$

$$SO_3 + H_2O \rightarrow 2\,H^+ + SO_4^{2-} \tag{9}$$

$$C_3AH_6 + 3\,Ca^+ + SO_4^{2-} + 26\,H \rightarrow 3\,CaO \cdot Al_2O_3 \cdot 3\,CaSO_4 \cdot 32\,H_2 \tag{10}$$

### 3.4.2. SS and CC Effects on the Pore Structure of UTCPB

The SEM-EDS testing method was used to determine the elemental composition of the surface and gel structure of ultra-fine tailings cemented paste backfill (UTCPB) specimens with and without admixture. Figure 10a depicts the microstructure of a UTCPB without admixture, and the surface of this specimen was magnified 10,000 times using an SEM. The ultra-fine tailing (UT) particles, the hydration products, and the formation of pores of various sizes between them can all be clearly seen in the SEM pictures. The quantity of hydration products was drastically decreased in comparison to Figure 9a because of the addition of UT particles. Between the tailing particles, crystallographic hydration products primarily take the form of a flocculated network. The flocculated network is inlaid with needles, plates, and other hydration products, which is the reason for the uniaxial compressive strength (UCS) increment in the UTCPB specimens. The microstructure of the UTCPB with admixture is shown in Figure 10c. It can also be clearly seen that micropores, needles, and amorphous gels are present in SEM images. While the hydration products form a denser structure with the UT particles in the admixture-added SEM image compared to Figure 10a without admixture. This is consistent with the pore characteristics of the two groups of specimens from the mercury intrusion porosimeter (MIP) experiment in Figure 11.

According to Figure 10b, the amorphous gel consists mainly of the elements O, Al, Si, and Ca by EDS tests. It was mutually confirmed that this gel structure was a silicate mineral by combining the XRD patterns of the net slurry hydration products of the cementitious materials and the EDX investigations. The elemental makeup of the amorphous gel structure of the ultra-fine tailings cemented paste backfill (UTCPB) samples with admixtures is depicted in Figure 10d. The main components of the gel structure within the UTCPB with admixtures did not change, but the elements S and Cl were discovered because chlorine was present in calcium chloride (CC). The generation of the element S was due to the dissociative effect on the blast furnace slag (BFS) in a strongly alkaline environment, leading to the production of $SO_4^{2-}$ and its participation in the hydration reaction of the BFS with carbide slag (CS). Additionally, the presence of the element Fe was discovered in Figure 10b, which may have resulted from tailing particles being mixed in with the gel structure because the tailings include a small amount of hematite.

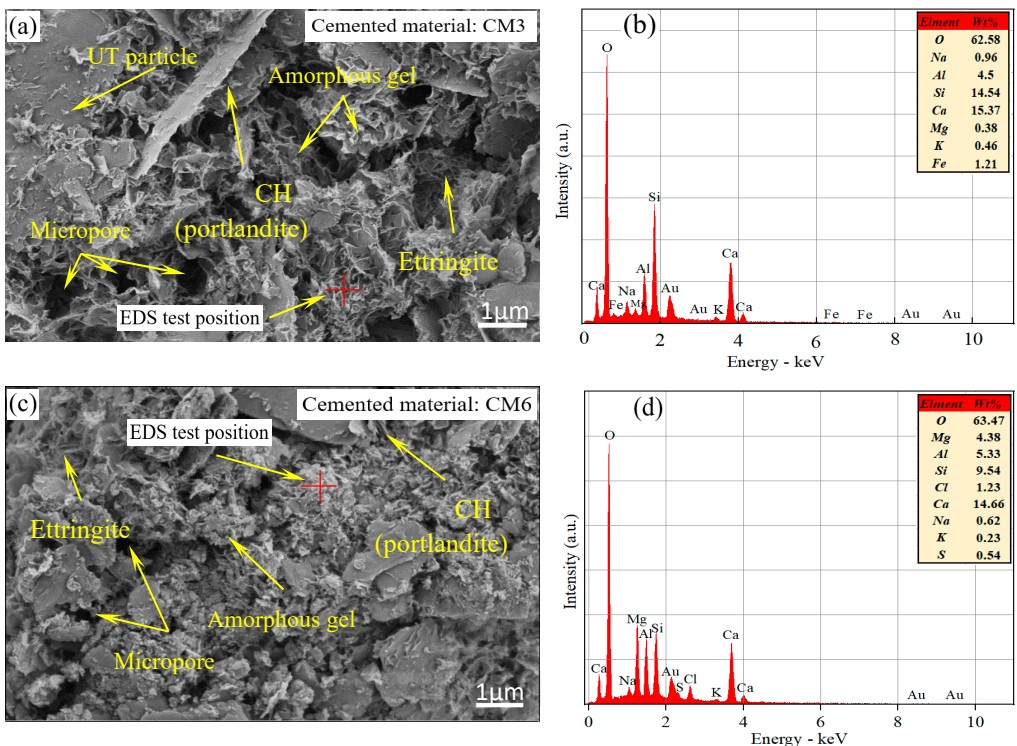

**Figure 10.** Microtesting of filling body specimens with a curing age of 7 days: (**a**) SEM images of UTCPB specimens without admixture; (**b**) elemental analyses from (**a**); (**c**) SEM images of UTCPB specimens with admixture; (**d**) elemental analyses from (**b**).

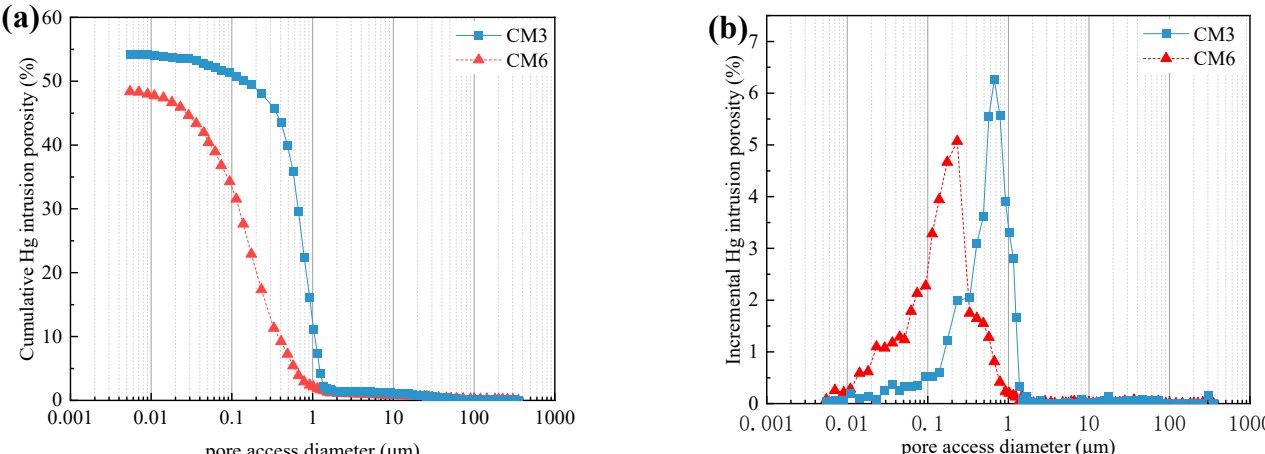

**Figure 11.** Influence of SS and CC on UTCPB: (**a**) cumulative intrusion curves; and (**b**) incremental volume curves.

### 3.4.3. SS and CC Effects on UTCPB Pore Size Distribution

Sodium silicate (SS) is a soluble inorganic silicate that can be classified as either a solid or a liquid, this test made use of liquid SS. SS has a wide range of uses, among which is its use as an admixture for cementitious materials to improve the stability and mechanical properties of concrete [42]. Ravikumar and Neithalath [43] pointed out that the addition of SS as an alkali excitant can cause the blast furnace slag (BFS) vitreous to further dissociate, and can increase the hydration reaction degree of BFS-based cementitious materials. The early generation of precipitates like $Ca(OH)_2$ and $CaSiO_3$ in the ultra-fine tailings cemented backfill slurry (UTCBS), which fill the internal pores and connect some of the tailing particles during the early consolidation of the ultra-fine tailings cemented

paste backfill (UTCPB), increased its early uniaxial compressive strength (UCS). Figure 11a shows that the porosity of the admixture-added UTCPB is lower than that of the admixture-unadded UTCPB. The total porosity of the admixture-added UTCPB is 48.37%, whereas the admixture-unadded UTCPB's total porosity is 54.25%, which is consistent with the experimental results of Section 3.4.2. For example, the age of curing is 7 days, the UCS of the admixture-added UTCPB is 2.76 MPa, and the UCS of the unadded UTCPB is 1.46 MPa. That shows admixture can decrease the internal porosity of UTCPB, increasing the microstructure's density. Additionally, the most probable pore size is used to describe the microstructure of the UTCPB. The most probable pore size of an admixture added is higher than that of an unadded one, as can be seen in Figure 11b, which was 0.23 μm (6.05%) with admixture, while it was 0.67 μm (7.12%) without admixture.

As a result, the porosity of the short-term ultra-fine tailings cemented paste backfill (UTCPB) decreased with the addition of admixture. The scanning electron microscopy (SEM) pictures in Figure 10 also show that the pore number and pore volume of the microstructure of the UTCPB without admixture were lower, and the surface of the tailings particles in the admixture-added ultra-fine tailings cemented backfill slurry (UTCBS) is coated with more hydration products like amorphous gel and ettringite. This reveals the intrinsic mechanism of the enhancement in the macroscopic mechanical characteristics of the admixture-added UTCPB.

*3.5. Cost Analysis of Cementitious Materials*

Traditional P. O32.5 and P. O42.5 cements are widely used for cementing tailings. However, the cement industry not only seriously pollutes the environment, but the instability of this market can also negatively affect mine production. According to China Cement (www.shuini.biz), it appears that ordinary Chinese P. O32.5 and P. O42.5 cements are currently purchased for RMB 290 and 330 yuan per ton, respectively, but in September 2019, the price soared to about RMB 500 yuan per ton. Conversely, using industrial by-products to make cementitious materials can lessen the environmental effects of industrial solid waste. At the same time, waste is turned into a valuable material, creating economic benefits. Blast furnace slag (BFS) is a by-product of the ironmaking process in blast furnaces, and the production of one ton of iron yields about 310 kg of BFS. So, it has a high yield. The addition of calcium carbide to water produces acetylene gas and carbide slag (CS), which is a solid waste with $Ca(OH)_2$ as its main component. At the same time, one ton of calcium carbide plus water produces about 1.2 tons of CS. The market survey indicates that the current prices per ton for BFS, CS, sodium silicate (SS), and calcium chloride (CC) are RMB 230, 140, 800, and 2000 yuan, respectively. Thus, S = 230 × 0.8 + 140 × 0.2 + 800 × 0.03 + 2000 × 0.01 = 256 yuan is the total cost per ton of the new cementitious material. Compared to ordinary P. O32.5 and P. O42.5 cements, this is substantially less expensive per ton.

**4. Conclusions**

In this paper, ultra-fine tailings (UT) with low comprehensive utilization rate were treated utilizing the underground mined-out areas cementation filling method, and to improve the cementing backfill effect of UT, the method of adding sodium silicate (SS) and calcium chloride (CC) admixtures in the ultra-fine tailings cemented backfill slurry (UTCBS) were investigated. The UTCBS was subjected to fluidity, setting time, and rheological tests, and the ultra-fine tailings cemented paste backfill (UTCPB) was subjected to uniaxial compressive strength (UCS) and microscopic performance tests. The main conclusions are as follows:

(1) A proper admixture dosage is beneficial to shorten the setting time of UTCBS, and the setting time of UTCBS in the blast furnace slag (BFS)-based cementitious system decreases as SS increases.

(2) UTCBS in the rheological experiment primarily manifests as "shear thinning" characteristics. At different shear rates, the shear stress-rate curve of UTCBS can be divided

into three stages: rapid growth, slow decline, and slow growth. As the hosting time increases, the second stage becomes more pronounced.

(3) The BFS-based cementitious material added with SS and CC significantly improved the short-term UCS of the UTCPB. When compared to P. O42.5 cement, the UCS of UTCPB made from BFS-based cementitious materials with admixture increased by 124%, 142%, and 13% at curing ages of 3, 7, and 28 days, respectively. Moreover, compared to ordinary P. O42.5 cement, this is less expensive at 22.4%.

(4) The BFS vitreous dissociates due to the strongly alkaline environment of the slurry produced by carbide slag (CS), producing C-S-H, C-A-H, $Ca(OH)_2$, a small amount of AFt, and other hydration products. O, Al, Si, and Ca are the main elements of the amorphous gel found in hydration products. With the addition of SS and CC, some calcium silicate precipitate was produced and wrapped around the UT particles, which caused the UCS of the UTCPB to significantly increase. The porosity of UTCPB without and with admixture at the curing age of 7 days was 54.25% and 48.37%, respectively.

Consequently, for better consolidation of UT, increasing the cement-tailing ratio will increase the cost of backfilling, and increasing the mass concentration will lead to transport difficulties. Although adding admixtures will somewhat increase the pipe transport resistance of UTCBS, using SS and CC in BFS-based cementitious materials for UT consolidation not only significantly reduces the cost of mine backfill, it also shortens the slurry's setting time and increases the UCS of UTCPB. As a result, this makes it gradually possible for ultra-fine tailings to backfill the mined-out areas. Finally, there is no need to construct a new tailings reservoir, minimizing the environmental impact of mining activity.

## 5. Limitations and Future Study

The work described in this article shows that the effect of adding water glass and calcium chloride on the rheological and consolidation properties of ultrafine tailing sand cemented filling slurries, is to greatly improve the early strength of ultrafine tailing sand filler, shorten the setting time, and reduce the cost of filling, although increasing the viscosity of the slurry to a certain extent. However, only one modulus of sodium silicate was considered, and it is hoped that the effects of multiple moduli of sodium silicate on the ultra-fine tailings backfill materials will be added to future research work. In addition, only calcium chloride was added, and it is hoped that the combination of other salts and sodium silicate will be added in future research work to enrich the research in the field of ultrafine tailing sand cementation. In addition, although the tailing sand cementation filling can solve the tailing sand surface stockpiling, it is undeniable that the slurry may have an impact on groundwater, and it is hoped that this part will be added to future research.

**Author Contributions:** Conceptualization, B.W. and S.G.; Data curation, L.Y.; Formal analysis, Z.W.; Funding acquisition, B.W.; Investigation, Z.Z. and J.W.; Methodology, S.G.; Project administration, S.G.; Resources, B.W.; Software, L.Y.; Supervision, B.W. and S.G.; Validation, Z.Z. and J.W.; Visualization, Z.Z.; Writing—original draft, S.G.; Writing—review & editing, Z.W. All authors have read and agreed to the published version of the manuscript.

**Funding:** This research was funded by the National Key Technologies Research & Development Program of China, grant number 2018YFC0808403.

**Data Availability Statement:** Full datasets are available from the corresponding author on request.

**Acknowledgments:** The authors are grateful to the journal editor and the anonymous reviewers for their constructive comments. We also thank Shandong Huayue Yitai Environmental Protection Science and Technology Co. for providing the platform for the experiment.

**Conflicts of Interest:** Zhongqi Zhao is employees of Shandong Gold Design Consulting Co., Ltd. The authors declare no conflicts of interest.

## Appendix A

**Table A1.** A list of acronyms.

| Acronyms | Full Names | Acronyms | Full Names |
|---|---|---|---|
| BFS | Blast furnace slag | CBS | Cemented backfill slurry |
| CC | Calcium chloride | CS | Carbide slag |
| C/T ratio | Cement-tailings ratio | CPB | Cemented paste backfill |
| MIP | Mercury intrusion porosimetry | SS | Sodium silicate |
| UCS | Uniaxial compressive strength | UT | Ultra-fine tailings |
| UTCBS | Ultra-fine tailings cemented backfill slurry | UTCPB | Ultra-fine tailings cemented paste backfill |
| SEM | Scanning electron microscopy | XRD | X-ray energy dispersive spectrometry |
| EDS | Energy dispersive spectrometer | - | - |

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
