# Peer review of "Additivity Effect on Properties of Cemented Ultra-Fine Tailings Backfill Containing Sodium Silicate and Calcium Chloride"

_minerals, doi:10.3390/min14020154_

Round 1
Reviewer 1 Report
Comments and Suggestions for Authors
General: Acronyms are used extensively in this paper. I needed to create a list of acronyms to keep readily at hand as I reviewed this paper. The list included:
BFS = blast furnace slag
CBS = cemented backfill slurry
CC = calcium chloride
CS = carbide slag
C/T ratio = cement-tailings ratio
CPB = cemented paste backfill
MIP = mercury intrusion porosimetry ???
SS = sodium silicate
UCS = uniaxial compressive strength
UT = ultra-fine tailings
UTCBS = ultra-fine cemented backfill slurry
UTCPB = ultra-fine cemented paste backfill
While the use of acronyms is widely practiced in scientific literature, for the sake of clarity, and to make reading the paper easier, I suggest that using the full term instead of the acronym is better, and does not add significant length to the paper. Alternatively, using acronyms in the same paragraph after use of the full term would also make reading much easier. There are times when brevity only adds to confusion.
Lines 20-23: The total percentages for "optimal proportioning" add up to more than 100%. I understand that materials are being added to the basic mixture in minor amounts, but totals need to add up to 100% in order to be rigorous in reporting the mixture percentages.
Lines 90-91: Reprocessing the tailings to produce post-tailings products that can be utilized has been very difficult to demonstrate in practice. The result is that few mines have been willing to attempt producing post-tailings products. The authors should address, or at least acknowledge, the fact that a market must exist for the various products produced from the raw tailings stream. If the markets are not willing to accept some or all of these post-tailings products, then that material will likely remain "tailings" that require disposal.
Line 206: Table 4. Specific parameters for each component of the cementitious materials
The totals in lines A4, A5, and A6 add up to more than 100%. I understand that Sodium Silicate and Calcium Chloride are being added to the basic mixture, but totals need to add up to 100% in order to be rigorous in reporting the mixture percentages.
Line 238: Typo: I assume the "S" in 120S-1 should be a lowercase "s"
Line 251: The acronym MIP needs to be defined.
Line 416; Can you confirm/explain what the abbreviations C, A, S, and H represent?
Line 456: Typo: cl = Cl?
Line 512: 4. Conclusions
The Uniaxial Compressive Strength increases with an increase in the Cement-Tailings ratio, Blast Furnace Slag - Carbide Slag & Sodium Silicate and Calcium Chloride, and Mass Concentration percentage. Can you opine any conclusions about what effect increasing these factors further would have, and whether further testing on these potential increases would be worthwhile?
Reviewer 2 Report
Comments and Suggestions for Authors
General comments on the manuscript:
In the manuscript titled 'Additivity Effect on Properties of Cemented Ultra-fine Tailings Paste Backfill Containing Sodium Silicate and Calcium Chloride', the authors investigate a cementitious material (cement backfill material) with proportioning parameters of 80% blast furnace slag, 20% carbide slag, and admixtures comprising 3% sodium silicate and 1% calcium chloride (the percentages of admixture are the ratios of the admixture to the total mass of blast furnace slag and carbide slag). The strength, rheology, and microstructure evolution of ultra-fine tailings cemented backfill slurry and cemented paste backfill were assessed through a series of macro and micro tests. The experimental results demonstrate that the short-term uniaxial compressive strength of ultra-fine tailings cemented paste backfill can be significantly enhanced by the newly formulated blast furnace slag-based cementitious material when compared to P.O 42.5 cement. The UCS of 3, 7, and 28 days can be improved by 124%, 142%, and 13%, respectively. The gold tailings are gradually becoming finer through the upgrading of beneficiation equipment and the increased comprehensive utilization rate of tailings, which significantly weakens the mechanical properties of the cemented paste backfill. This study serves as a valuable reference for the practical application of gold ultra-fine tailings cemented backfill, both technically and economically.
The manuscript is interesting. It is relatively scientifically novel. Cement backfill is an interesting and attractive area of research due to the constant request for circularity. The research is scientifically sound. The text is highly readable and understandable. Generally, this manuscript looks good with sufficient experimental techniques. However, there are still some shortcomings in the manuscript, and it is recommended that it be revised for publication.
Specific comments:
(1) The title does not describe the research correctly. The description in the article refers to "cemented ultra-fine tailings backfill" rather than " cemented ultra-fine tailings paste backfill", as the term "paste" is inaccurately used. Authors should consider changing it.
(2) In Table 5, there is an error in the expression of the formula for Mass concentration. Please verify.
(3) The abstract is informative. Authors should avoid using abbreviations in the abstract as much as it is possible.
(4) The list of keywords should be improved and more specific to describe the research. Please do not repeat terms already used in the title.
(5) Please mention the limitations of this study.
(6) Fig. 8: Please provide the standard deviations in the column chart.
(7) Results are elaborately explained and discussed. All provided figures in this section are necessary and of good quality. There is no need for condensation of text.
(8) References are up-to-date.
(9) Conclusions are adequate as they are following the provided and discussed results. At the end of the Conclusion section, the authors are suggested to present the future studies. Such as, “And, the potential pollution of backfill materials to underground water will also be further studied in the future, which has been a hot topic in mine backfill [ref.s].
[ref.1] Yikai Liu, Yunming Wang, et al. Using cemented paste backfill to tackle the phosphogypsum stockpile in China: A down-to-earth technology with new vitalities in pollutants retention and CO2 abatement, Int. J. Miner. Metall. Mater.,(2023). https://doi.org/10.1007/s12613-023-2799-y],
[ref.2] Immobilization and leaching characteristics of fluoride from phosphogypsum-based cemented paste backfill, Int. J. Miner. Metall. Mater., 28(2021), No. 9, pp. 1440-1452. https://doi.org/10.1007/s12613-021-2274-6].]”
(10) The English language is correct and understandable. Authors should re-read the text one more time to remove all potential typing and formatting errors.
Comments on the Quality of English Language
None
Reviewer 3 Report
Comments and Suggestions for Authors
The paper on 'Additivity Effect on Properties of Cemented Ultra-fine Tailings Paste Backfill Containing Sodium Silicate and Calcium Chloride' studied the effects of sodium silicate (SS) and calcium chloride (CC) on the properties of cementitious ultra-fine tailings filling materials.The paper is well organized, the writing is correct. Before publication, Authors should make a revision considering the review comments listed below. The Reviewer believes that the raised comments will be easily addressed by the authors, and an updated version can be provided for publication.
1. It is mentioned in the abstract and conclusion that the compressive strength of UTCPB material at 3 days, 7 days and 27 days increased by 124%, 142% and 13% respectively, which data are not found in the text, please add it in time in 3.3. In addition, 13% of the calculation results are wrong, please check the calculation results.
2. The abstract and introduction are too long, and the author is advised to simplify the language. Problem statements, requirements, and solutions should be carefully reflected.
3. The novelty of the research should be emphasized more clearly at the end of the introduction. How does this study differ from published studies in the literature?
4. The content of calcium chloride in the article is 1%, please explain the reason for choosing its content. Does the amount of calcium chloride affect the final result?
5. In article 3.4.1, the introduction of chemical reactions does not correspond to the serial number of the formula. Also, where is the formula 10 mentioned in the article? Please check it carefully.
6. How will the results of this study benefit researchers and end users? This point needs to be emphasized in the concluding remarks.
I would be happy to see the revised version to understand how these comments are being addressed.
